# Oral Health and Frailty in Community-Dwelling Older Adults in the Northern Netherlands: A Cross-Sectional Study

**DOI:** 10.3390/ijerph19137654

**Published:** 2022-06-23

**Authors:** Coen Dros, Martine J. Sealy, Wim P. Krijnen, Lina F. Weening-Verbree, Hans Hobbelen, Harriët Jager-Wittenaar

**Affiliations:** 1Research Group Healthy Ageing, Allied Health Care and Nursing, Hanze University of Applied Sciences, Petrus Driessenstraat 3, 9714 CA Groningen, The Netherlands; coendros1992@hotmail.com (C.D.); w.p.krijnen@pl.hanze.nl (W.P.K.); l.f.weening-verbree@pl.hanze.nl (L.F.W.-V.); j.s.m.hobbelen@pl.hanze.nl (H.H.); ha.jager@pl.hanze.nl (H.J.-W.); 2FAITH Research, Petrus Driessenstraat 3, 9714 CA Groningen, The Netherlands; 3Johan Bernoulli Institute for Mathematics and Computer Science, University of Groningen, 9700 AK Groningen, The Netherlands; 4Center for Dentistry and Oral Hygiene, University Medical Center Groningen, Antonius Deusinglaan 1, FB 21, 9713 AV Groningen, The Netherlands; 5Department of General Practice and Elderly Care Medicine, University of Groningen, University Medical Center Groningen, Hanzeplein 1, 9713 GZ Groningen, The Netherlands; 6Department of Oral and Maxillofacial Surgery, University of Groningen, University Medical Center Groningen, Hanzeplein 1, 9713 GZ Groningen, The Netherlands

**Keywords:** oral health, frailty, GFI, older adults, OHIP, healthy ageing, ageing

## Abstract

The aim of this study was to explore the association between oral health and frailty in community-dwelling Dutch adults aged 55 years and older. Included were 170 participants (*n* = 95 female [56%]; median age 64 years [IQR: 59–69 years]). Frailty was assessed by the Groningen Frailty Indicator. Oral health was assessed by the Oral Health Impact Profile-14-NL (OHIP-NL14). OHIP-NL14 item scores were analyzed for differences between frail and non-frail participants. Univariate and multivariate logistic regression analyses were performed to assess the association between oral health and presence of frailty. The multivariate analysis included age, gender, and depressive symptoms as co-variables. After adjustment, 1 point increase on the OHIP-NL14 scale was associated with 21% higher odds of being frail (*p* = 0.000). In addition, significantly more frail participants reported presence of problems on each OHIP-NL14 item, compared to non-frail participants (*p* < 0.003). Contrast in prevalence of different oral health problems between frail and non-frail was most prominent in ‘younger’ older adults aged 55–64 years. In conclusion: decreased oral health was associated with frailty in older adults aged ≥55 years. Since oral health problems are not included in most frailty assessments, tackling oral health problems may not be sufficiently emphasized in frailty policies.

## 1. Introduction

The global population is ageing. Currently, almost every country in the world is experiencing a growth in the proportion of older adults in the population [1]. The increase in the number of older adults is accompanied by an increase in the prevalence of frailty [2]. While the current prevalence of frailty in community-dwelling older adults in The Netherlands is estimated to be approximately 16% [3,4]; this is projected to be about 25% in 2030 [5].

Frailty is a dynamic state affecting an individual who experiences losses in one or more domains of human functioning (physical, psychological, social), which is caused by the influence of a range of variables and which increases the risk of adverse outcomes [6]. While frailty has a considerable negative impact on the quality of life of older adults [7] and may result in a 1.8 to 2.3-fold increased risk for mortality [8], frailty is also potentially reversible [6]. Mounting evidence suggests that early identification and intervention slow down the progression of frailty or even prevent it [9]. Longitudinal data show that in community-dwelling older adults, a transition from a frail to a non-frail state is possible. However, not all factors associated with this transition have been examined [10]. Therefore, identifying all factors associated with frailty transition and early identification of frailty is urgently needed. If multiple physical, psychological, and social problems of older adults are identified as early as possible, treatment could be optimized and adverse health effects could be prevented [11].

A specific, yet important, health problem associated with ageing is poor oral health [12]. The World Dental Federation (FDI) defines oral health as follows: “oral health is multi-faceted and includes the ability to speak, smile, smell, taste, touch, chew, swallow and convey a range of emotions through facial expressions with confidence and without pain, discomfort, and disease of the craniofacial complex”. Previous research has shown that approximately one in three older adults report that their oral health is poor [13]. Therefore, in recent years, the role of prevention within oral healthcare is being emphasized [14]. Poor oral health has been associated with diabetes risk, cardiovascular disease, and respiratory diseases [15]. For example, periodontitis is an inflammatory disease of the oral mucosa, where dysbiosis plays an important role, that adversely impacts systemic health [16,17]. Worldwide, oral health problems are highly prevalent in older adults, and with the world’s population ageing, the cumulative burden of oral conditions largely increased between 1990 and 2015 [18]. Therefore, it is important to consider oral health as a factor associated with healthy ageing.

A systematic review on the relationship between oral health and frailty showed that poor oral health is a predictor of frailty onset. However, the studies included in the review predominantly focused on physical frailty and older adults aged 65 years or older, resulting in inclusion of older populations with a mean age of 70+ years [19,20]. Therefore, it is unclear if oral health and frailty are associated in older adults at a younger age (i.e., >55 years). Insight into the relationship between oral health and frailty in “younger” older adults will strengthen the possibility for early identification of risk of frailty onset and may enable preventive measures. Therefore, in this study, we aimed to explore the association between oral health and frailty in community-dwelling adults aged 55 years and older in the Northern Netherlands.

## 2. Materials and Methods

To test the hypothesis that oral health is associated with frailty, a cross-sectional analysis of data collected in the Hanze Health and Ageing Study (HHAS) was performed. The HHAS is conducted by the Research Group Healthy Ageing, Allied Health Care and Nursing at the Hanze University of Applied Sciences, Groningen, The Netherlands. The Medical Ethical Committee of the University Medical Center Groningen (UMCG) approved the HHAS, under the regulation of the Medical Research Involving Human Subjects Act (reference METc 2016/446).

### 2.1. Study Population

The population of this cross-sectional study consists of community-dwelling older adults (≥55 years of age) who live in the Northern Netherlands. We based the sample size estimation for sufficient power on the following information: average frailty incidence of participants from the Northern Netherlands was 9% as measured with the GFI [21]. In edentulous persons from the Northern Netherlands, the prevalence of frailty was 26% [22]. This group was 75 years and older, and not all people with decreased oral health are edentulous. Therefore, we assumed that theaverage incidence of frailty in persons with oral health problems was 16% instead of 26%. If the chance of a type I error is set at 0.05 and the chance of a type II error at 0.20, a sample size of 154 is estimated to provide sufficient power for our calculations. Subjects who participated in the HHAS between September 2016 and February 2020 were included. Participants were recruited through advertising in newspapers, social media, community centers, and from social networks. The participants were informed about the HHAS procedures and gave their informed consent prior to participating in this study.

### 2.2. Measurements

#### 2.2.1. Frailty

Frailty was assessed by the Groningen Frailty Indicator (GFI), which is based on multiple domains in which pathways are described on how the loss of functioning through physical, psychological, cognitive, and social domains affects health [23]. The GFI questionnaire consists of 15 items with the possibility to score 1 point per item with a total of 15. In item one to four, ‘no’ = 1 point. In item five to eight, ‘yes’ = 1 point. Item nine rates physical fitness with a scale from 0 to 10 (0–6 = 1 point; 7–10 = 0 points). In item 11–15, ‘yes’ = 1 point, ‘sometimes’ = 1 point, and ‘no’ = 0 points. A total GFI score ≥4 points was used as cut-off to indicate frailty [24]. See Appendix A.

#### 2.2.2. Oral Health

Oral health was assessed by the Oral Health Impact Profile-14-NL (OHIP-NL14). The OHIP-NL14 was validated in older adults with a mean age of <60 years. Furthermore, higher mean total OHIP-NL14 scores were associated with lower self-reported oral health status, more dentures, having fewer teeth, more dental complaints, more dental complaints related to disability [25], more swallowing problems, and overall poor self-perceived oral health [26]. The questionnaire consists of 14 items. The answers to the 14 items were scored on ordinal scales, ranging from: ‘never’ (0 points), ‘rarely’ (1 point), ‘occasionally’ (2 points), ‘rather often’ (3 points), to ‘very often’ (4 points). The 14 scale scores were then summed; the total score can thus range from 0 to 56 points. See Appendix A.

#### 2.2.3. Co-Variables

A wide range of variables that are known to be associated with both oral health and frailty were included in the multivariate analysis [18,27]. The following variables were considered potentially associated: age, gender, and educational status, cognitive status, depressive symptoms, body mass index (BMI), comorbidity, and smoking status.

The demographic variables age (years), gender (categories: female or male), and education status (categories: low, middle, or high) were self-reported. Education was categorized as low (education level: primary school not completed, primary school, (pre-)vocational education), middle (secondary school and secondary vocational education), or high (higher general and/or preparatory education, higher professional education, university education).

Cognitive status was assessed by the Qmci-D. The Qmci-D consists of six components, i.e., orientation, registration, clock drawing, delayed recall, verbal fluency, and logical memory, whereby a total of 100 points can be obtained (scale: 0–100). No clinical cut-off values are set for Qmci-D. However, lower scores have been associated with cognitive impairment [28].

The Center of Epidemiological Studies Depression Scale-Dutch (CES-D) questionnaire was completed by participants as a raw indicator for clinical depression. The CES-D consists of 20 multiple-choice questions. Each question consists of 4 answering options: <1 day (0 points), 1–2 days (1 point), 3–4 days (2 points), or 5–7 days (3 points), whereby a maximum of 3 points per question can be obtained, resulting in a total of 60 points (scale: 0–60). Participants scoring ≥16 points are considered as ‘possible clinically depressed’ [29].

Comorbidity was retrospectively assessed by the Age-adjusted Charlson Comorbidity Index (ACCI) [30]. The ACCI considers the following 19 medical conditions: myocardial infarction, angina, cardiovascular disease, cerebrovascular disease, dementia, chronic obstructive pulmonary disease (COPD), connective tissue disease, gastrointestinal disease, slight or serious liver disease, diabetes mellitus complicated, stroke, kidney failure, cancer, leukemia, lymphoma, secondary metastasis, and AIDS. In addition, age is also scored. The score ranges from 1 to 6 for each item, and the total score provides an index of severity of comorbidity. Subjects who scored ≥4 points were indicated as having comorbidity.

Furthermore, body mass index (BMI) (kg/m^2^) was calculated by correcting weight in kilograms measured with an electronic scale (Seca 761, Seca, Hamburg Germany) for stature in meters measured with a tape measure (Seca 201, Seca, Hamburg, Germany). Finally, smoking status (categories: current smoker, ex-smoker, or non-smoker) was self-reported with ‘yes’, ‘no, I quitted smoking’ or ‘no, never smoked’.

### 2.3. Data Analysis

Descriptive statistics were used to provide an overview of the characteristics of the study participants and, based on their characteristics, presented with absolute numbers and percentages for ordinal and dichotomous variables. Furthermore, mean and standard deviation, or median scores and interquartile ranges, are reported for continuous variables. In addition, descriptive statistics were used to assess the frequencies of the OHIP-NL14 scores and the differences in proportions of OHIP-NL14 scores ≥1 in frail vs. non-frail participants overall and after stratifying for age groups 55–64 years and ≥65 years. A Fisher exact test was performed on the dichotomized OHIP-NL14 score (0 = no and 1 to 4 = yes). Since the OHIP-14 is comprised of 14 items, we applied a Bonferroni correction for multiple comparisons for the analysis on item level, for which the level of significance was set at *p* < 0.003.

A univariate binary logistic regression analysis was performed to assess the association between self-reported oral health, the presence of frailty, and all the above-mentioned potential interaction variables. The univariate analysis without correction was followed by a multivariate binary logistic regression analysis adjusted for potential interaction variables. In addition, the multivariate logistic approach was followed by a Smoothy Clipped Absolute Deviation (SCAD) penalty approach to select contributing variables without creating bias. For selecting the explanatory variables, the value of the penalty parameter is determined by the corrected Akaike information criterion [31,32]. Finally, a multivariate binary logistic regression analysis was performed to assess the association between the OHIP-NL14 and presence of frailty, corrected for the selected contributing variables. For a few subjects with missing observations for the variables education status (*n* = 1) and cognitive function (*n* = 6), data imputation was used according to the Multivariate Imputation by Chained Equations procedure. All statistical analyses were performed with R (version: 4.0.0, R Core Team Vienna, 201X) and the level of significance was set at *p* < 0.05.

## 3. Results

### 3.1. Sample Characteristics

The HHAS data included survey responses from 181 participants, of which 11 (6%) were excluded due to missing frailty data. Of the remaining 170 participants, the median age was 64 years (IQR: 59–69) and 95 participants were female (56%). The participants had an overall median score of 0 (IQR: 0–2) on the OHIP-NL14 and 1 (IQR: 0–2) on the GFI. In total, 37 (22%) were frail. Non-frail participants had a median GFI score of 1 (IQR: 0–2) and frail participants had a median GFI score of 5 (IQR: 4–6). The baseline characteristics of participants are presented in Table 1.

Overall, participants most frequently scored ‘never’ on items on the OHIP-NL14, ranging from 58% who scored ‘never’ for oral pain (item 3), to 95% who scored ‘never’ for difficulty doing your job (item 12). Thus, oral pain (item 3) was the most prevalent oral health problem, and difficulty doing your job was the least prevalent oral health problem. Frequency of OHIP-14-NL items’ scores is reported in Table 2.

### 3.2. Oral Health

On each OHIP-NL14 item, except item 3, i.e., aching pain in the mouth, significantly more frail participants reported problems compared to non-frail participants (*p* < 0.003). Table 3 shows the differences in scores between frail and non-frail participants for each OHIP-NL14 item. For both frail (*n* = 22; 60%) and non-frail participants (*n* = 49; 37%), item 3, i.e., aching pain in the mouth, was the most prevalent problem reported.

The relative difference in OHIP-NL item scores ≥1 of frail vs. non-frail participants aged 55–64 years and for frail vs. non-frail participants aged ≥65 years is presented in Figure 1. Within the age category 55–64 years (*n* = 100), frail participants (*n* = 18; 18%) significantly more often rated ≥1 for 8/14 OHIP-NL14 items (*p* < 0.003). These eight items were:Item 2, i.e., sense of taste worsened;Item 6, i.e., tense feeling;Item 8, i.e., interrupt meals because of problems;Item 9, i.e., difficult to relax;Item 11, i.e., irritable with other people;Item 12, i.e., difficulty doing your usual jobs;Item 13, i.e., life in general less satisfying;Item 14, i.e., totally unable to function.

Within the age category ≥65 years (*n* = 70), frail participants (*n* = 19; 27%) significantly more often (*p* < 0.003) rated ≥1 on the OHIP-NL14 for item 4, i.e., found it uncomfortable to eat any foods.

### 3.3. Oral Health and Frailty

As shown in Table 4, in the univariate logistic regression analysis, OHIP-NL14 was significantly associated with the odds of being frail (RC: 0.22, *p* < 0.05). In addition, a significant association was found between the co-variables age, cognitive function, depressive symptoms, and frailty. After correction for all potential interaction variables, a higher score on the OHIP-NL14 was still significantly associated with higher odds of being frail (RC: 0.22, *p* < 0.002, r^2^ = 0.54). Finally, the SCAD analysis selected oral health, age, gender, and depressive symptoms explaining half of the GFI variance by a r^2^ of 0.50, as shown in Table 5. This shows that the variables selected by the SCAD selection explain a large amount of the variation in the GFI score when compared to the model with all interaction variables. After correction for age, gender, and depressive symptoms, the OHIP-NL14 was associated with an approximately 21% increase in the odds of being frail (OR: 1.21 95% CI: 1.09–1.37).

## 4. Discussion

In this study, poor oral health was associated with the presence of frailty in community-dwelling older adults aged 55 years and older. An increase in one unit on the OHIP-NL14 scale was associated with 21% higher odds of being frail after correction for age, gender, and depressive symptoms. In the age group 55–64 years, reported oral health problems in frail older adults contrasted more clearly with non-frail peers than in the age group ≥65 years.

Our study includes relatively young older adults aged 55–64 years. However, the finding that oral health was associated with frailty is consistent with studies in older populations (median age 71–83 years). In studies that also assessed frailty with the GFI, frail older adults were more often edentulous and reported more oral problems than non-frail older adults [22,33,34].

A novel finding of our study is that in the group of ‘younger’ older adults, oral health problems of the frail persons contrasted more clearly with those of their non-frail peers than in the group of ‘older’ older adults. This indicates the contrast in prevalence of different oral health problems between frail and non-frail was most prominent in ‘younger’ older adults aged 55–64 years. In contrast to ‘younger’ frail older adults, ‘older’ frail older adults did not report feeling that life in general was less satisfying or that they were unable to function because of problems with their teeth, mouth, or dentures. These results may suggest that oral health does not affect the quality of life of frail ‘older’ older adults as much as in ‘younger’ older adults. This inverse association between age and self-reported oral health, also referred to as the disability paradox, has been reported in previous studies [35]. The disability paradox of better subjective oral health in older age states that while oral health problems occur more frequently in older age, older adults of 66 years of age and older may hold a positive perception of oral health and have adapted to tooth loss [26,35,36]. A difference in oral health problems between age groups may infer that there is a need for oral health strategies focused on older adults in an early stage, to prevent a negative impact on quality of life. With regard to frailty, this may imply a need for oral health strategies in an early stage of oral health problems to prevent frailty.

Due to the multidimensional nature of oral health and frailty, both physical and psychosocial pathways can explain the association between oral health and frailty. In the present study, frail older adults experienced more physical oral health problems, such as aching pain in the mouth, problems with eating, and had to change their diet due to problems with their mouth, teeth, and dentures. These findings are consistent with previous research showing the interplay between oral health problems and nutrition [37,38]. Physical discomfort and problems with eating due to problems with mouth, teeth, and dentures may impact food choices, which can lead to a decrease in nutritional status both quantitatively (lower energy and protein intake) and qualitatively (lower fiber and vitamin intake) [37]. Considering that nutritional status may play an important role in the development of frailty, having physical oral health problems can strengthen the onset and development of frailty [38]. Furthermore, in our study, frail older adults also experienced more psychosocial oral health problems, such as having been self-conscious, having felt tense, having been embarrassed, and having felt that life in general was less satisfying due to problems with their teeth, mouth, or dentures. These associations are consistent with previous research showing that poor oral health can result in frailty through impairing self-esteem and social interactions leading to social isolation, which is an independent risk factor for the progression of frailty [39,40].

### 4.1. Strengths and Limitations

A strength of this study is that we were able to include a broad range of demographic, lifestyle, and health-related co-variables in the multivariate analysis, thus correcting for a wide range of factors that may potentially interact with oral health and/or frailty.

This study also has some limitations. Firstly, the sample was somewhat underrepresented by older adults with a lower education, current smokers, and older adults with comorbidities. Because these factors are associated with negative health outcomes, the association between oral health and frailty could be underestimated. Secondly, even though the OHIP-NL14 is associated with numerous oral health indicators, this study did not include a physical assessment of the dental status by an oral health care professional and oral health was self-reported. However, it has been reported that oral health assessed by dental health care professionals often agrees with self-reported oral health [26,41]. Considering the OHIP-NL14 as a measurement for oral health allows for a shift from traditional medical/dental criteria, such as teeth count or oral pain, to an oral health assessment that focuses on a person’s social and emotional experience and physical functioning. Thirdly, the association between oral health and frailty may have a bidirectional nature, since the association between aspects of frailty such as functional disability or depression could also be related to a decline in oral health [42,43]. Finally, this study did not consider oral frailty, which is a relatively new concept related to oral health. Oral frailty is a stage prior to oral disfunction. Oral frailty has been defined as a decrease in oral function together with a decline in cognitive and physical functions and has been recognized and related to the broader concept of frailty [19,44,45]. By addressing oral frailty, attention will most likely be drawn to oral dysfunction instead of dysfunction in broader domains of human functioning (physical, psychological, social) [10]. Therefore, in this study, we focused on the association between oral health and frailty.

### 4.2. Implications of Findings for Research and Practice

At present, oral health items are not included in most frailty assessment instruments. Therefore, tackling oral health problems may not be sufficiently emphasized in frailty policies. Both oral health and frailty are potentially preventable or reversible and attention for oral health problems may at least partially prevent frailty. Therefore, the findings of our study have implications for practice, in particular from a public health perspective. Policies and strategies must be strengthened to prevent both oral health problems and frailty. Strategies should take into account age, in particular because oral health problems are potentially more easily treated in ‘younger’ older adults. Since the frequency of dental check-up visits is declining, actively encouraging older adults to have regular check-ups may potentially prevent oral health problems and frailty, since problems may be identified at an earlier stage [46]. Strategies aimed at ‘younger’ older adults may also focus on more psychosocial problems such as improving self-esteem and preventing social isolation. However, due to the underlying problems with teeth, mouth, or dentures as root causes of the psychosocial and physical problems, education should also focus on oral hygiene such as regular oral health assessments and the use of the correct teeth-cleaning tools to improve overall oral health [47]. Furthermore, further research should consider oral health as a factor associated with the transition between frail and non-frail status. Preventive strategies may need to include the multidimensional aspects of oral health, i.e., physical or psychosocial problems, when targeting frailty in order to be effective.

## 5. Conclusions

Decreased oral health was firmly associated with frailty in community-dwelling Dutch older adults aged ≥55 years. Since oral health problems are not included in most frailty assessments, tackling oral health problems may not be sufficiently emphasized in frailty policies. The contrast in prevalence of different oral health problems between frail and non-frail was most prominent in ‘younger’ older adults aged 55–64 years. Therefore, preventive policies could target ‘younger’ older adults.

## Figures and Tables

**Figure 1 ijerph-19-07654-f001:**
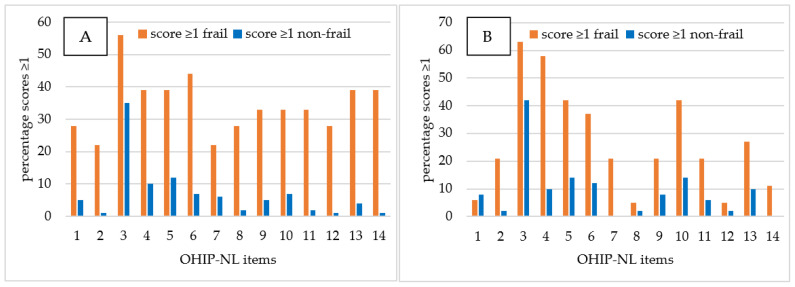
Percentages of OHIP-NL item scores ≥1 of frail (*n* = 18) vs. non-frail (*n* = 82) participants aged 55–64 years (**A**) and frail (*n* = 19) vs. non-frail (*n* = 51) participants aged ≥65 years (**B**).

**Table 1 ijerph-19-07654-t001:** Baseline characteristics of participants overall and by frailty status.

Characteristics		Total	Frail	Non-Frail
		*n* = 170 (100)	*n* = 37 (22)	*n* = 133 (78)
Frailty (GFI; median (IQR))		1 (0–2)	5 (4–6)	1 (0–2)
Oral health (OHIP-NL14; median (IQR))		1 (0–2)	3 (0–12)	2 (0–2)
Age (in years; median (IQR))		64 (59–69)	64 (59–69)	66 (62–71)
Gender (%)	Male	75 (44)	14 (38)	61 (46)
	Female	95 (56)	23 (62)	72 (54)
Education status (%)	Low	14 (8.2)	2 (5)	12 (9)
	Middle	49 (29)	7 (19)	42 (32)
	High	107 (63)	28 (76)	79 (59)
Smoking status (%)	Non-smoker	69 (41)	12 (32)	57 (43)
	Current smoker	8 (5)	1 (3)	7 (5)
	Ex-smoker	93 (55)	24 (65)	69 (52)
Body mass index (kg/m^2^; median (IQR))		25 (23–28)	25 (23–27)	25 (23–28)
Cognitive function (Qmci-D; mean (SD))		70 (10)	69 (10)	71 (10)
Depressive symptoms (CES-D; median (IQR))		6 (2–11)	14 (9–20)	4 (1–8)
Comorbidity (%)	No	155 (91)	32 (87)	123 (93)
	Yes	15 (9)	5 (14)	10 (8)

Standard deviation = SD, Interquartile range = IQR.

**Table 2 ijerph-19-07654-t002:** Frequency of Oral Health Impact Profile-14-NL (OHIP-NL14) items scores (*n* = 170).

		OHIP-NL14 Score				
Item	Question	Never *n* (%)	Rarely *n* (%)	Occasionally *n* (%)	Rather Often *n* (%)	Very Often *n* (%)
1	Have you had trouble pronouncing certain words because of problems with your teeth, mouth or dentures?	151 (89)	11 (6)	5 (3)	2 (1)	1 (1)
2	Have you felt that your sense of taste has worsened because of problems with your teeth, mouth or dentures?	160 (94)	8 (5)	2 (1)	0 (0)	0 (0)
3	Have you had pain aching in your mouth?	99 (58)	52 (31)	15 (9)	4 (2)	0 (0)
4	Have you found it uncomfortable to eat any foods because of problems with your teeth, mouth or dentures?	139 (82)	24 (14)	4 (2)	3 (2)	0 (0)
5	Have you been self-conscious because of your teeth, mouth or dentures?	138 (81)	18 (11)	12 (7)	2 (1)	0 (0)
6	Have you felt tense because of your teeth, mouth or dentures?	143 (84)	19 (11)	8 (5)	0 (0)	0 (0)
7	Has your diet been unsatisfactory because of your teeth, mouth or dentures?	157 (92)	11 (6)	1 (1)	1 (1)	0 (0)
8	Have you had to interrupt meals because of problems with your teeth, mouth or dentures?	161 (95)	7 (4)	2 (1)	0 (0)	0 (0)
9	Have you found it difficult to relax because of problems with your teeth, mouth or dentures?	152 (89)	15 (9)	2 (1)	1 (1)	0 (0)
10	Have you been a bit embarrassed because of problems with your teeth, mouth or dentures?	143 (84)	18 (10)	8 (5)	1 (1)	0 (0)
11	Have you been a bit irritable with other people because of problems with your teeth, mouth or dentures?	155 (91)	12 (7)	3 (2)	0 (0)	0 (0)
12	Have you had difficulty doing your usual jobs because of problems with your teeth, mouth or dentures?	162 (95)	7 (4)	0 (0)	1 (1)	0 (0)
13	Have you felt that life in general was less satisfying because of problems with your teeth, mouth or dentures?	150 (88)	13 (8)	5 (3)	2 (1)	0 (0)
14	Have you been totally unable to function because of your problems with your teeth, mouth or dentures?	160 (94)	9 (5)	1 (1)	0 (0)	0 (0)

**Table 3 ijerph-19-07654-t003:** Differences in scoring for each OHIP-NL14 item between frail and non-frail community-dwelling older adults.

	Frail	Non-Frail	Frail vs. Non-Frail
OHIP-NL14 Item	*n* (%)	*n* (%)	*p*-Value
1	11 (30)	8 (6)	0.000 *
2	8 (22)	2 (2)	0.000 *
3	22 (60)	49 (37)	0.015
4	18 (49)	13 (10)	0.000 *
5	15 (41)	17 (13)	0.001 *
6	15 (41)	12 (9)	0.000 *
7	8 (22)	5 (4)	0.001 *
8	6 (16)	3 (2)	0.000 *
9	10 (27)	8 (6)	0.001 *
10	14 (38)	13 (10)	0.000 *
11	10 (27)	5 (4)	0.000 *
12	6 (16)	2 (2)	0.001 *
13	12 (32)	8 (6)	0.000 *
14	9 (24)	1 (1)	0.000 *

* *p*-value after applying the Bonferroni correction for multiple comparisons for the analysis on item level, level of significance was calculated at *p* < 0.003.

**Table 4 ijerph-19-07654-t004:** Univariate, multivariate, and SCAD analysis for the association between oral health and frailty.

		Univariate Analysis		Multivariate Analysis		SCAD Analysis
		RC ^1^	*p*-Value	RC ^1^	*p*-Value	RC ^1^
Oral health (OHIP-NL14)		0.22	0.000 *	0.22	0.002 *	0.12
Age (in years)		0.06	0.021 *	0.06	0.241	0.02
Female		0.33	0.386	1.49	0.014 *	1.14
Education status	High					
	Low	−0.52	0.507	−1.46	0.177	-
	Middle	−0.65	0.159	−1.17	0.074	-
Smoking status	Non-smoker					
	Current smoker	−0.69	0.523	−1.20	0.431	-
	Ex-smoker	0.53	0.163	0.31	0.564	-
Body mass index		0.00	0.969	0.04	0.423	-
Cognitive function (Qmci-D)		−0.04	0.041 *	−0.04	0.183	-
Depressive symptoms (CES-D)		0.20	0.000 *	0.18	0.000 *	0.18
Comorbidity		0.65	0.262	−0.51	0.651	-

^1^ Regression coefficient, * Significance was set at *p* < 0.05.

**Table 5 ijerph-19-07654-t005:** Univariate and multivariate logistic analyses after SCAD selection of variables associated with frailty.

	Univariate Analysis			Multivariate Analysis		
	OR ^1^	95% CI ^2^	*p*-Value	OR ^1^	95% CI ^2^	*p*-Value
Oral health (OHIP-NL14)	1.25	1.13–1.38	0.000 *	1.21	1.09–1.37	0.000 *
Age (in years)	1.06	1.01–1.12	0.021 *	1.06	0.99–1.15	0.086
Female	1.39	0.66–2.94	0.386	3.26	1.16–10.44	0.032 *
Depressive symptoms (CES-D)	1.22	1.13–1.31	0.000 *	1.20	1.12–1.31	0.000 *

^1^ Odds ratios, ^2^ 95% confidence intervals, * Significance was set at *p* < 0.05.

## Data Availability

The data presented in this study are not publicly available. Data can be made available upon request from the corresponding author.

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
