# Peer review of "Oral Health and Frailty in Community-Dwelling Older Adults in the Northern Netherlands: A Cross-Sectional Study"

_ijerph, 2022, doi:10.3390/ijerph19137654_

Round 1
Reviewer 1 Report
The authors present a cross-sectional study to explore the association between oral health and frailty in community-dwelling adults aged 55 year and older.
The aim is relevant for preventive policy needs in dentistry and public health fields. The manuscript is clear, relevant for the field and presented in a well-structured manner. The cited references are mainly within the last 5 years.
Little editing of English language is required;
Reference list should corrected (line 378 to 468) to e ACS style guide. Please see instructions for authors. Example : “line 378- “published online “ should be removed.
Title- Should be re-wrote in order to not be un affirmative conclusion. I suggest to short the title to the main purpose, without the “affirmative sentence” “oral health is associated to…”Please uniformize title.
Abstract – Suggestion avoid unnecessary repetition, as “line 21 “- Remove “ “older” before “aged 55 years and older”.
Main: Lines 110, 113 and 115 – Suggest to correct “are” to “were”
Results chapter- Include description of “Table 1” in text – between lines 185 and 191.
Discussion chapter- Suggestion to correct “is” to “was” in line 267 and other parts of text.
We-Wrote to better clarify and synthetise the sentence lines315 -316; and also line 329;
Conclusion- Suggestion to correct “is” to “was” – line 348.
Author Response
Thank you very much for giving us the opportunity to revise our manuscript. We have revised our manuscript according to the valuable comments of the reviewers. The changes are recognizable in the manuscript by track changes and the corresponding line numbers in the marked version of the manuscript are listed below.
Reviewer 1
The authors present a cross-sectional study to explore the association between oral health and frailty in community-dwelling adults aged 55 year and older.
The aim is relevant for preventive policy needs in dentistry and public health fields. The manuscript is clear, relevant for the field and presented in a well-structured manner. The cited references are mainly within the last 5 years.
Little editing of English language is required;
- Reference list should corrected (line 378 to 468) to e ACS style guide. Please see instructions for authors. Example : “line 378- “published online “ should be removed.
We thank the reviewer for this suggestion and we have now corrected the reference list according to the ACS style guide.
- Title- Should be re-wrote in order to not be un affirmative conclusion. I suggest to short the title to the main purpose, without the “affirmative sentence” “oral health is associated to…”Please uniformize title.
To avoid stating an affirmative conclusion, the title has been adapted as follows: Oral health and frailty in community dwelling older adults in the Northern Netherlands: A cross-sectional study
- Abstract – Suggestion avoid unnecessary repetition, as “line 21 “- Remove “ “older” before “aged 55 years and older”.
We thank the reviewer for this suggestion and we have removed “older” from the sentence in line 21.
- Main: Lines 110, 113 and 115 – Suggest to correct “are” to “were”
We thank the reviewer for pointing this out and we have corrected “are” to “were”, as suggested in lines 123 , 127 and 129 .
- Results chapter- Include description of “Table 1” in text – between lines 185 and 191.
We thank the reviewer for this suggestion and we have now included a description of “Table 1” in the text in lines 205-206.
- Discussion chapter- Suggestion to correct “is” to “was” in line 267 and other parts of text.
We thank the reviewer for pointing this out and we have corrected “is” to “was” as suggested.
- We-Wrote to better clarify and synthetise the sentence lines315 -316; and also line 329;
We agree with the reviewer and we have now re-written the sentence in line 332 to 334 as follows: “A strength of this study is that we were able to include a broad range of demographic, lifestyle, and health-related co-variables in the multivariate analysis, thus correcting for a wide range of factors that may potentially interact with oral health and/or frailty.”
The sentence in lines 358 to 362 has now been rephrased as follows: “Even though, oral health is associated to frailty, oral health items are not included in most frailty assessment instruments. Therefore, the importance of oral health problems may be underemphasized in frailty policies. Both oral health and frailty are potentially preventable or reversable and attention for oral health problems may at least partially prevent frailty. Therefore, the findings of our study have implications for practice, in particular from a public health perspective.
- Conclusion- Suggestion to correct “is” to “was” – line 348.
We thank the reviewer for pointing this out and we have corrected “is” to “was”, as suggested in line 379 .
Reviewer 2 Report
Manuscript of considerable interest, before being published it needs a major revision.
Abstract: statistically significant data to be added
Keywords: to add others, these are few
Introduction: add the incidence of periodontal and mucosal disease according to the age group, insert the assessment and training methods also in dental hygiene and dentistry courses, as illustrated in the research group of Prof. Scribante et al.
Materials and methods: how was the sample size calculated? What home oral hygiene aids did they use? how were they educated?
Results. statically significant data in the tables should be highlighted
Discussion: to be added as a future objective to maintain a state of eubiosis over time, the use of probiotics, paraprobiotics and postbiotics for the maintenance of periodontal and microbiological indices, widely described by the research group of Prof. Scribante.
Conclusions: add proactive action
Bibliography: add the required references.
Author Response
Thank you very much for giving us the opportunity to revise our manuscript. We have revised our manuscript according to the valuable comments of the reviewers. The changes are recognizable in the manuscript by track changes and the corresponding line numbers in the marked version of the manuscript are listed below.
Reviewer 2
Manuscript of considerable interest, before being published it needs a major revision.
- Abstract: statistically significant data to be added.
We thank the reviewer for pointing this out and we have now added the statistically significant data to the abstract.
- Keywords: to add others, these are few
We have now added the following keywords: OHIP; healthy ageing; ageing
- Introduction: add the incidence of periodontal and mucosal disease according to the age group, insert the assessment and training methods also in dental hygiene and dentistry courses, as illustrated in the research group of Prof. Scribante et al.
We agree that the incidence of periodontal and mucosal disease is of importance in this age group and that periodontal health affects general health. However, in this study we have chosen to focus on oral health impact as reported by older people. The focus of our study fits well in the definition of oral health as stated by The World Dental Federation (FDI): “oral health is multi‐faceted and includes the ability to speak, smile, smell, taste, touch, chew, swallow and convey a range of emotions through facial expressions with confidence and without pain, discomfort, and disease of the craniofacial complex.”
We also agree that periodontal and mucosal disease may be important factors in the mechanism that may explain the association between oral health and frailty. We have now described periodontitis as a possible pathway for the association between oral health and chronic disorders, such as diabetes and cardiovascular disease, in lines 68-70 .
- Materials and methods: how was the sample size calculated?
We calculated the sample size for sufficient power as follows: average frailty incidence of participants from the Northern Netherlands was 9% as measured with the GFI (Peters et al, 2015). In edentulous persons from the Northern Netherlands, the prevalence of frailty was 26% (Hoeksema et al. 2017). This group was 75 years and older, and not all people with decreased oral health are edentulous. Therefore, we corrected the incidence of frailty in persons with oral health problems from 26% to 16%. If the chance of a type I error is set at 0.05 and the chance of a type II error at 0.20, a sample size of 154 is estimated to provide sufficient power for our calculations. This is now stated in the Methods section in lines 95 to 102.
- Materials and methods: What home oral hygiene aids did they use? how were they educated?
In this observational study, we focused on the broader concept of oral health and did not collect information regarding oral hygiene habits of people participating in our study and participants were not educated.
- Statically significant data in the tables should be highlighted.
Thank you for pointing this out. We have now highlighted the significant results in the table 3, 4 and 5.
- Discussion: to be added as a future objective to maintain a state of eubiosis over time, the use of probiotics, paraprobiotics and postbiotics for the maintenance of periodontal and microbiological indices, widely described by the research group of Prof. Scribante.
Although we think eubiosis is very important to maintain a healthy periodontal environment, our study focuses on the broader concept of oral health defined by the FDI as follows: oral health is multi‐faceted and includes the ability to speak, smile, smell, taste, touch, chew, swallow and convey a range of emotions through facial expressions with confidence and without pain, discomfort, and disease of the craniofacial complex. We do not aim to focus on the specific biological and physiological aspects of oral flora; instead we aimed to provide more insight into what the relationship between oral health and frailty implies for the community and for health care workers in general. Therefore, in our opinion this objective does not fit within the focus of the current study.
- Conclusions: add proactive action.
We thank the reviewer for this suggestion and we have now revised the conclusion to be more proactive, in line 380 to 384.
- Bibliography: add the required references.
As a result of the revisions that we made, we have now added the following references:
Hajishengallis, G; Chavakis, T. Local and systemic mechanisms linking periodontal disease and inflammatory comorbidities. Nat Rev Immunol. 2021, 21, 426–440.
Butera A; Gallo S; Pascadopoli M; Maiorani C; Milone A; Alovisi M; et al. Paraprobiotics in Non-Surgical Periodontal Therapy: Clinical and Microbiological Aspects in a 6-Month Follow-Up Domiciliary Protocol for Oral Hygiene. Microorganisms. 2022,10, 337.
Reviewer 3 Report
The findings of the present study seem to be clear and well documented.
However, some “limitations” should be discussed:
Apart from oral health problems, frailty could be due to the general fatigue of some aged people or/and to the depression of some others. Regarding the opposite way of the results’ evaluation, oral health would not be a priority for a fatigued or/and depressed individual, so in such cases oral health problems could be the result and not the cause of frailty.
Furthermore, if similar questionnaires were distributed to similar populations, similar results might have been concluded regarding other health issues, besides oral ones, as well. Findings of relevant studies could be reported.
Author Response
Thank you very much for giving us the opportunity to revise our manuscript. We have revised our manuscript according to the valuable comments of the reviewers. The changes are recognizable in the manuscript by track changes and the corresponding line numbers in the marked version of the manuscript are listed below.
Reviewer 3
The findings of the present study seem to be clear and well documented.
However, some “limitations” should be discussed:
- Apart from oral health problems, frailty could be due to the general fatigue of some aged people or/and to the depression of some others. Regarding the opposite way of the results’ evaluation, oral health would not be a priority for a fatigued or/and depressed individual, so in such cases oral health problems could be the result and not the cause of frailty.
We agree with the reviewer that the relationship between frailty and oral health may be bi-directional. Therefore, we have now described this in the limitations, in lines 346 to 348. “Thirdly, the association between oral health and frailty may have a bidirectional nature, since the association between aspects of frailty such as functional disability or depression could also be related to decline of oral health.” (Avlund K et al., 2001. Cademartori, M, et al., 2018)
- Furthermore, if similar questionnaires were distributed to similar populations, similar results might have been concluded regarding other health issues, besides oral ones, as well. Findings of relevant studies could be reported.
The association between health related issues such as depressive symptoms, cognitive problems and co-morbidity burden and frailty has been demonstrated in a large body of evidence. Psychological and cognitive problems, as well as multimorbidity, are concepts included in the multidimensional construct of frailty. Therefore, these concepts are incorporated in most multidimensional frailty assessment instruments, such as the GFI. However, the concept of oral health is not tested for in most frailty assessment instruments. Therefore, we think it is of extra importance to draw attention to the association between oral health and frailty. To emphasize this importance, we have now added the following text to the implications section lines 358 to 360:
“At present, oral health items are not included in most frailty assessment instruments. Therefore, tackling oral health problems may be not sufficiently emphasized in frailty policies.”
Reviewer 4 Report
In the current demographic context, the subject approached by the authors is very topical.
The association of oral health with aging is of great interest to specialists in all medical fields.
I note, however, that the authors did not address the issue of oral fragility in the study, a stage prior to oral hypofunction.
„The oral frailty phenotype is a novel construct proposed as a conceptualisation of age-related gradual loss of oral function, driven by a set of impairments that worsen oral daily functions—eg, loss of teeth, poor oral hygiene, inadequate dental prostheses, or difficulty in chewing associated with age-related changes in swallowing. Oral frailty has been defined as a decrease in oral function together with a decline in cognitive and physical functions, such as oral microbiota and Alzheimer's disease neurodegeneration”.
Vittorio Dibello, Roberta Zupo, Rodolfo Sardone, Madia Lozupone, Fabio Castellana, Antonio Dibello, Antonio Daniele, Giovanni De Pergola, Ilaria Bortone, Luisa Lampignano, Gianluigi Giannelli, Francesco Panza, Oral frailty and its determinants in older age: a systematic review, The Lancet Healthy Longevity, Volume 2, Issue 8, 2021,Pages e507-e520:
Other publications addressing oral frailty:
Tanaka T, Hirano H, Ohara Y, Nishimoto M, Iijima K. Oral Frailty Index-8 in the risk assessment of new-onset oral frailty and functional disability among community-dwelling older adults. Arch Gerontol Geriatr. 2021 May-Jun;94:104340
Tanaka T, Takahashi K, Hirano H, Kikutani T, Watanabe Y, Ohara Y, Furuya H, Tetsuo T, Akishita M, Iijima K. Oral Frailty as a Risk Factor for Physical Frailty and Mortality in Community-Dwelling Elderly. J Gerontol A Biol Sci Med Sci. 2018 Nov 10;73(12):1661-1667.
Takeuchi, Noriko et al. “Oral Factors as Predictors of Frailty in Community-Dwelling Older People: A Prospective Cohort Study.” International journal of environmental research and public health vol. 19,3 1145. 20 Jan. 2022,
Ayoob AK, Neelamana SK, Janakiram C. Impact of oral frailty on general frailty in geriatric population: A scoping review. J Indian Assoc Public Health Dent 2022;20:9-15
Being a newer concept, I think it should be mentioned at least in the chapter „Discussions”.
The conclusions need to be rephrased because they do not relate to the results obtained in the study.
Author Response
Thank you very much for giving us the opportunity to revise our manuscript. We have revised our manuscript according to the valuable comments of the reviewers. The changes are recognizable in the manuscript by track changes and the corresponding line numbers in the marked version of the manuscript are listed below.
Reviewer 4
In the current demographic context, the subject approached by the authors is very topical.
The association of oral health with aging is of great interest to specialists in all medical fields.
- I note, however, that the authors did not address the issue of oral fragility in the study, a stage prior to oral hypofunction.
„The oral frailty phenotype is a novel construct proposed as a conceptualisation of age-related gradual loss of oral function, driven by a set of impairments that worsen oral daily functions—eg, loss of teeth, poor oral hygiene, inadequate dental prostheses, or difficulty in chewing associated with age-related changes in swallowing. Oral frailty has been defined as a decrease in oral function together with a decline in cognitive and physical functions, such as oral microbiota and Alzheimer's disease neurodegeneration”.
Vittorio Dibello, Roberta Zupo, Rodolfo Sardone, Madia Lozupone, Fabio Castellana, Antonio Dibello, Antonio Daniele, Giovanni De Pergola, Ilaria Bortone, Luisa Lampignano, Gianluigi Giannelli, Francesco Panza, Oral frailty and its determinants in older age: a systematic review, The Lancet Healthy Longevity, Volume 2, Issue 8, 2021,Pages e507-e520:
Other publications addressing oral frailty:
Tanaka T, Hirano H, Ohara Y, Nishimoto M, Iijima K. Oral Frailty Index-8 in the risk assessment of new-onset oral frailty and functional disability among community-dwelling older adults. Arch Gerontol Geriatr. 2021 May-Jun;94:104340
Tanaka T, Takahashi K, Hirano H, Kikutani T, Watanabe Y, Ohara Y, Furuya H, Tetsuo T, Akishita M, Iijima K. Oral Frailty as a Risk Factor for Physical Frailty and Mortality in Community-Dwelling Elderly. J Gerontol A Biol Sci Med Sci. 2018 Nov 10;73(12):1661-1667.
Takeuchi, Noriko et al. “Oral Factors as Predictors of Frailty in Community-Dwelling Older People: A Prospective Cohort Study.” International journal of environmental research and public health vol. 19,3 1145. 20 Jan. 2022,
Ayoob AK, Neelamana SK, Janakiram C. Impact of oral frailty on general frailty in geriatric population: A scoping review. J Indian Assoc Public Health Dent 2022;20:9-15
Being a newer concept, I think it should be mentioned at least in the chapter „Discussions”.
We thank the reviewer for this suggestion and we agree that the concept of ‘oral frailty’ is of great interest for early detection of oral problems. However there is also a risk towards specifically focussing on oral frailty, since attention will most likely be drawn to oral aspects of frailty and problems in other frailty domains may be overlooked. We have now added a paragraph in the Strengths and Limitations section to address oral frailty in lines 348 to 355. ‘Finally, this study did not consider oral frailty, which is a relatively new concept related to oral health. Oral frailty is a stage prior to oral disfunction. Oral frailty has been defined as a decrease in oral function together with a decline in cognitive and physical functions and has been recognized and related to the broader concept of frailty [19, 44, 45]. By addressing oral frailty, attention will most likely be drawn to oral disfunction instead of disfunction in broader domains of human functioning (physical, psychological, social) [10]. Therefore, in this study we focused on the association between oral health and frailty.’ (Parisius et al, 2022; Tanaka et al. 2018; Dibello et al, 2021)
- The conclusions need to be rephrased because they do not relate to the results obtained in the study.
We have now rephrased the conclusions as follows in lines 379-384:
‘Decreased oral health was firmly associated with frailty in community-dwelling Dutch older adults aged ≥55 years. Since oral health problems are not included in most frailty assessments, tackling oral health problems may be not sufficiently emphasized in frailty policies. Contrast in prevalence of different oral health problems between frail and non-frail was most prominent in ‘younger’ older adults aged 55-64 years. Therefore, preventive policy could target ‘younger’ older adults.’
Round 2
Reviewer 2 Report
The manuscript has been correctly revised, it can be published